# Stabilization of body balance with Light Touch following a mechanical perturbation: Adaption of sway and disruption of right posterior parietal cortex by cTBS

**David Kaulmann**[1]*, **Matteo Saveriano**[2], **Dongheui Lee**[3,4], **Joachim Hermsdörfer**[1], **Leif Johannsen**[5]

1 Department of Sport and Health Sciences, Human Movement Science, Technische Universität München, Munchen, Germany, 2 Department of Computer Science, Intelligent and Interactive Systems, University of Innsbruck, Innsbruck, Austria, 3 Human-centered Assistive Robotics, Electro- and Information Technology, Technische Universität München, Munchen, Germany, 4 Institute of Robotics and Mechatronics, German Aerospace Centre, Cologne, Germany, 5 Institute of Psychology, Cognitive and Experimental Psychology, RWTH Aachen, Aachen, Germany

* david.kaulmann@tum.de

**Data Availability Statement:** The experimental Data is accessible via the institutional media repository of the Technical University Munich:

## Abstract

Light touch with an earth-fixed reference point improves balance during quite standing. In our current study, we implemented a paradigm to assess the effects of disrupting the right posterior parietal cortex on dynamic stabilization of body sway with and without Light Touch after a graded, unpredictable mechanical perturbation. We hypothesized that the benefit of Light Touch would be amplified in the more dynamic context of an external perturbation, reducing body sway and muscle activations before, at and after a perturbation. Furthermore, we expected sway stabilization would be impaired following disruption of the right Posterior Parietal Cortex as a result of increased postural stiffness. Thirteen young adults stood blindfolded in Tandem-Romberg stance on a force plate and were required either to keep light fingertip contact to an earth-fixed reference point or to stand without fingertip contact. During every trial, a robotic arm pushed a participant's right shoulder in medio-lateral direction. The testing consisted of 4 blocks before TMS stimulation and 8 blocks after, which alternated between Light Touch and No Touch conditions. In summary, we found a strong effect of Light Touch, which resulted in improved stability following a perturbation. Light Touch decreased the immediate sway response, steady state sway following re-stabilization, as well as muscle activity of the Tibialis Anterior. Furthermore, we saw gradual decrease of muscle activity over time, which indicates an adaptive process following exposure to repetitive trials of perturbations. We were not able to confirm our hypothesis that disruption of the rPPC leads to increased postural stiffness. However, after disruption of the rPPC, muscle activity of the Tibialis Anterior is decreased more compared to sham. We conclude that rPPC disruption enhanced the intra-session adaptation to the disturbing effects of the perturbation.

https://mediatum.ub.tum.de/1500480?show_id=1546513.

**Funding:** We acknowledge the financial support by the Federal Ministry of Education and Research of Germany (BMBF; 01EO1401) and by the Deutsche Forschungsgemeinschaft (DFG) through the TUM International Graduate School of Science and Engineering (IGSSE). The funders had no role in study design, data collection and analysis, decision to publish, or preparation of the manuscript.

**Competing interests:** The authors have declared that no competing interests exist.

**Abbreviations:** CNS, Central Neuro System; CoG, Centre of Gravity; CoM, Centre of Mass; CoP, Centre of Pressure; cTBS, continuous Theta Burst Stimulation; dCoP, differentiated Centre of Pressure; IPG, Inferior Parietal Gyrus; LT, Light Touch; PPC, Posterior Parietal Cortex; rTMS, repetitive Transcranial Magnetic Stimulation; rPPC, right Posterior Parietal Cortex; TMS, Transcranial Magnetic Stimulation.

## Introduction

The main objective for the control of body posture and balance is to stabilize upright standing against the pull of gravity or any other external forces and to prevent the body from toppling over. This is achieved by keeping the Centre of Mass' (COM) vertical projection onto the ground (Centre of Gravity, CoG) within the support boundaries. In order to maintain balance, the Central Nervous System (CNS) relies on sensory feedback processed by the visual, vestibular and somatosensory systems [1]. However, in addition to its primary senses the CNS is also able to use information from secondary afferent channels, such as the skin, as long sway-related information is conveyed. Light touch (LT) with an earth-fixed reference point has been shown to decrease sway variability and improve balance during quite stance [2] but also in dynamic situations, such as when compensating an either foreseeable or unpredictable external perturbation. For example, Dickstein and colleagues [3] demonstrated that Light Touch facilitates the scaling of postural compensation in response to horizontal support surface translations. Furthermore, Light Touch results in faster stabilization and reduced body sway following both externally and self-imposed body balance perturbations [4]. Imposing the sudden release of a backward load to the trunk, Martinelli et al. [5] reported that Light Touch reduced and slowed Centre-of-Pressure (CoP) displacement as well as decreased activity in the lower limbs' Gastrocnemius muscles under challenging sensory conditions. Johannsen and co-workers [6] also provided evidence for the benefit of Light Touch in dynamic postural contexts by exerting abrupt backward perturbations onto participants standing on a compliant springboard under different conditions of visual feedback. The utilization of Light Touch stabilized balance and decreased thigh muscle activity by up to 30%, which indicates that Light Touch optimizes mechanical and metabolic costs of balance compensation following a perturbation to a compliant support surface [6].

Although responses to postural perturbations are faster than voluntary movements, the observation that long-latency reflexes are sensitive to the postural context suggests involvement of supraspinal neural circuits including the cerebral cortex [7]. Several studies implied a role of cortical neural circuits in the control of posture when anticipating a perturbation to body balance. Cortical potentials preceding self-initiated perturbations, as well as predictable external perturbations show differences in amplitude as well as temporal characteristics [8], which might represent adjustments in a central set prior to the onset of a known perturbation. Depending on alterations in the cognitive state, such as changes in the cognitive load or attentional focus, initial sensory-motor conditions, prior experience and prior warning of a perturbation influences the central set enabling adaptations of the postural response to a perturbation [7]. Several cortical areas have been identified for playing a role in the control of balance, mainly the primary motor cortex, the somatosensory cortex and the posterior parietal cortex (PPC). For example, the primary motor cortex is responsible in the regulation of induced postural responses of the lower limbs [9]. Taube et al. [9] applied a single pulse TMS paradigm to demonstrate that corticospinal projection to the soleus muscle facilitates long-latency responses following abrupt backward translations of the support. Similarly, the sensorimotor cortex has been reported to play a role not only in the integration and in processing of sensory information, but also in adjusting the central set to modify externally triggered postural responses [7]. In addition, involvement of the supplementary motor area in motor planning and preparation for an adequate response to perturbations has been reported [10–12]. Contrasting balance perturbations caused by horizontal translations of a support surface with and without an auditory pre-warning, Mihara et al. [10] used functional near-infrared spectroscopy to demonstrate that both the left-hemisphere supplementary motor area and the right-hemisphere posterior parietal cortex increased activation, when preparation for the upcoming perturbation was possible. This observation argues for an involvement of both areas

in the anticipation and probably also compensation of an expected postural imbalance. Likewise, An et al. [13] who investigated the contribution of the sensory motor cortex and the PPC to recovery responses following unpredictable perturbations during standing or walking. Both areas showed a suppressed activity in the alpha band during periods of balance recovery [13]. The significant role of the posterior parietal cortex in the stabilization of balance is further corroborated by Lin et al. [14]. They showed that a lesion in the posterior parietal cortex following stroke leads to reactive postural control deficit, such as impaired recruitment of paretic leg muscles and a more frequent occurrence of compensatory muscle activation patterns compared to controls. Lin et al. [14] concluded that the PPC is part of a neural circuitry involved in reactive postural control in response to lateral perturbations.

Regions of the cerebral cortex are also involved in the processing and integration of the sensory information from the fingertips when utilizing Light Touch for postural control. Ishigaki et al. [15] demonstrated involvement of the left primary sensorimotor cortex and the left posterior parietal cortex in stance control with light tactile feedback. Johannsen et al. [16] investigated how rTMS over the left inferior parietal gyrus (IPG) influences sensory re-organization for the control of postural sway with light fingertip contact. They reported that rTMS over the left IPG reduced overshoot of sway after contact removal, which indicates that this brain region may play a role in inter-sensory conflict resolution and adjustment of a central postural set for sway control with contralateral fingertip contact.

Assuming that an ego-centric reference frame would be the basis of interpreting and disambiguating fingertip Light Touch for sway control in a quiet upright stance with transitions between postural states with and without Light Touch feedback, we investigated the effects of disrupting the left- and right hemisphere PPC using continuous Theta Burst Stimulation (cTBS) [17]. We expected that disruption of the right Posterior Parietal Cortex would impair integration of Light Touch into the postural control loop and attenuate the effect of Light Touch on body sway. These expectations were not confirmed but we demonstrated that rPPC disruption influenced the complexity of body sway with Light Touch of the non-dominant, contralateral hand [17]. In addition, disruption of the rPPC resulted in an overall sway reduction and altered complexity irrespective of the presence of Light Touch. A possible reason could be that rPPC disruption increased overall body stiffness due to lower limb muscular co-contractions and thus reduced body sway [18]. Sway reduction does not mean, however, that participants are intrinsically more stable. Variability is a means of the postural control system to achieve a specific task goal while at the same time being more able to react flexibly to possible external balance perturbations [19]. Thus, it can be argued that the reduction in sway reflects an unfavourable effect in terms of participants becoming less adaptive and less able to compensate unexpected perturbations [20] after rPPC disruption.

Taking into account the well documented light-touch-related facilitation of balance stabilization, following an external perturbation [3,4,5,6] we implemented a perturbation paradigm to assess the influence of rPPC disruption on dynamic stabilization of body sway with and without Light Touch. In previous studies, however, perturbations consisted either of a single constant force or of variable forces but in a blocked design, making perturbations much more predictable, enabling adjustment to a central postural set. In our current study, we intended to make it much more difficult for the participants to predict the force of an upcoming perturbation. Therefore, we randomized three forces on a trial-by-trial basis within a block of either Light Touch or no touch. We hypothesized that the benefit of Light Touch would be amplified in the more dynamic context of an external perturbation to balance, improving the compensation response. We also expected that the immediate response to a perturbation and sway stabilization in terms of its time constant would be affected expressing an increase in postural stiffness following rPPC disruption.

## Methods

### Participants

Thirteen healthy right-handed young adults (age = 26 ± 2 (SD); 10 women and 3 men) were recruited for this study, using the faculties own blackboard. Inclusion criteria were (1) no neurological or musculoskeletal disorders, (2) no balance impairment and (3) no known history of epilepsy or reported seizures. All participants were informed about the study protocol and signed a written informed consent. The study was approved by the Clinical Research Ethics committee of the Medical School of the Technical University Munich.

### Study protocol, apparatus and experimental procedure

The study protocol comprised of two single TMS sessions in the balance lab. The order of stimulation locations (rPPC or sham TMS) was randomized across participants. Stimulation sessions were separated by at least 24 hours. Each experimental testing session consisted of three parts: a balance pre-test, 60 seconds of cTBS and a balance post-test. During the pre- and post-test participants stood in Tandem-Romberg stance on a force plate (600Hz; Bertec FP4060-10, Columbus, Ohio, USA), with their eyes blindfolded and instructed to stand quietly but relaxed and not to attempt to minimize body sway.

Participants were required either to keep light haptic fingertip contact with their dominant hand to an earth-fixed reference point or to stand without fingertip contact. Participants practiced keeping Light Touch with the reference point prior to the start of the experiment receiving verbal feedback about the strength of the contact force until they felt comfortable maintaining Light Touch below 1 N. During the experiment, however, participants did not receive feedback about contact force to prevent contacting from becoming an explicit, attention-demanding precision task. The earth-fixed contact reference point was placed in front of the participants. They held one arm slightly angled in front of the body and reaching straight forward. The other arm remained passive at the side of their body (Fig 1).

Body kinematics (4 Oqus 500 infrared cameras; 120 Hz; Qualisys, Göteborg, Sweden) and forces and torques at the fingertip reference contact location (6DoF Nano 17 force-torque transducer; 200 Hz; ATI Industrial Automation, Apex, USA) were also acquired. To capture body motion, reflective markers were placed at the contacting fingertip, wrist, elbows, shoulders, C7, Sternum, hip, knees and ankles. Additionally, surface EMG (1kHz) of the Gastrocnemius, Soleus and Tibialis Anterior of the posterior supporting leg was recorded to measure muscle activity (Trigno Wireless PM-W05, Delsys, Natic, MA, USA).

During every single standing trial, a robotic arm (KUKA LBR4+, Augsburg, Germany) exerted a push to participants at their right shoulder in medio-lateral direction. In order to make the next perturbation force as unpredictable as possible, the force of a lateral push was exerted with either 1%, 4% or 7% of their respective body weight in a randomized order in a block consisting of 6 trials (2 trials for each push force). Using a percentage of the body weight for every single participant, results in different absolute forces for the participants. However, relative force of the push for the perturbation is equalized for across participants. Table 1 shows the absolute peak push forces in N for the conditions averaged over all participants.

A testing session consisted of 4 blocks before the cTBS application (pre-test) and 8 blocks after (post-test). The blocks alternated between Light Touch (LT) and No Touch (NT) conditions. For a comparison between sway before and after the cTBS application, sway was averaged across the NT and LT blocks respectively (pre-test: NT = blocks 1+3, LT = blocks 2+4; post-test: NT = blocks 6+8+10+12; LT = blocks 5+7+9+11). Duration of a single trial was 20 seconds, with the lateral push always applied at 4.5 seconds after the start of a trial (Fig 2).

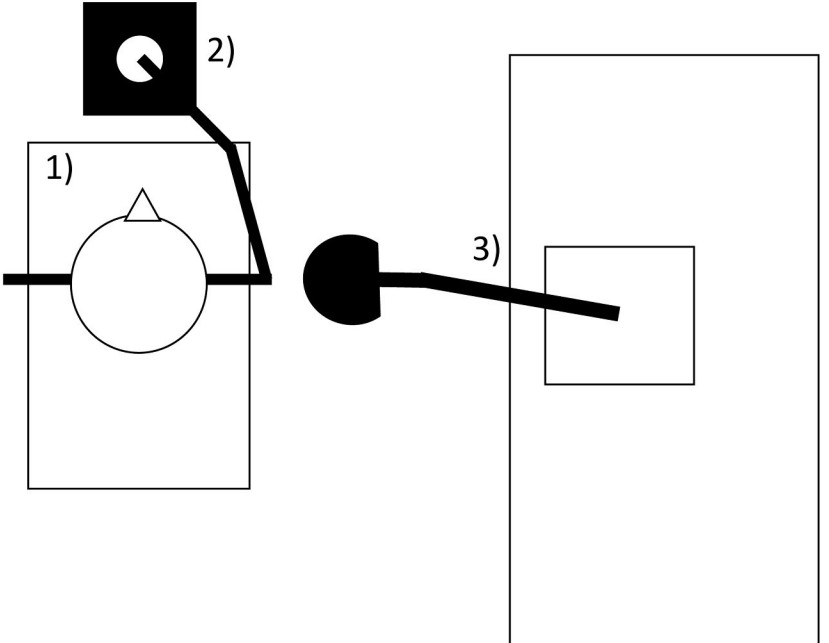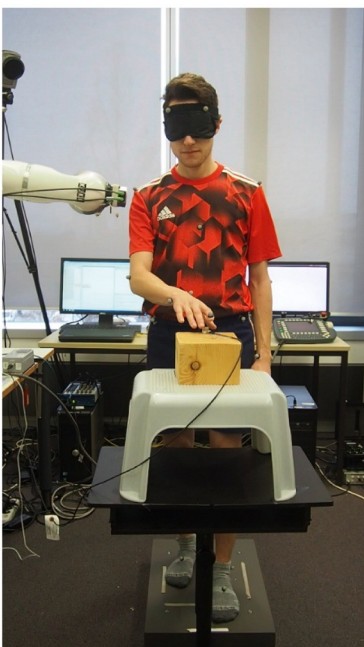

**Fig 1. Experimental set up as seen from above.** (1) Force plate, (2) contact reference point on a waist high stand and (3) Robotic arm mounted on a table.

## Neuronavigation and TMS protocol

During cTBS stimulation, participants were seated comfortably on a reclined chair facing a wall and keeping their head straight. We applied continuous Theta Burst Stimulation (cTBS) of an intensity of 80% of the passive motor threshold for 60 seconds over the rPPC (PMD70-pCool; MAG & More, Munich, Germany). This protocol is widely used and stimulation effects can last from 20 minutes up to 1 hour (Staines & Bolton [21]. The passive motor threshold was determined by registering the motor evoked potential (MEP) at the musculi interossei dorsales manus of the left hand following a single TMS pulse over the hand representation of the right-hemisphere primary motor cortex. A staircase procedure was used to adjust the pulse intensity until a 50μV MEP could be elicited reliably [22].

Sham stimulation was applied over the same target location as for the cTBS using a sham coil powered at similar intensities, which produced no focussed magnetic induction but created similar acoustics and tactile sensation. (PMD70-pCool-Sham; MAG & More, Munich, Germany).

High-resolution anatomical brain scans were acquired before the study at the University Hospital Großhadern, Center for Sensorimotor Research and consisted of a T1 MPRAGE (3T

**Table 1. Push forces averaged over all participants broken down by force push condition and stimulation protocol.**

| % of Body Weight | Stimulation Protocol | Force (N) |
|---|---|---|
| 1 | Sham | 2.99 |
| 1 | Stim | 2.89 |
| 4 | Sham | 6.95 |
| 4 | Stim | 6.01 |
| 7 | Sham | 11.56 |
| 7 | Stim | 10.06 |

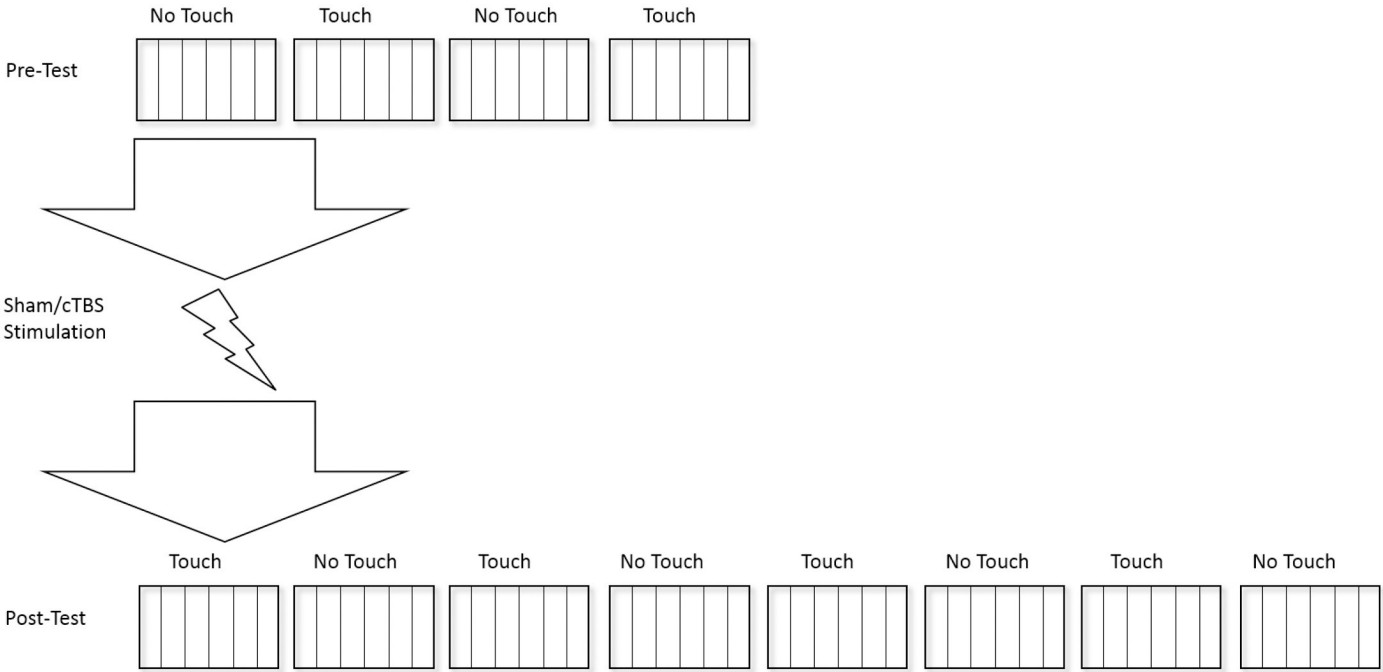

**Fig 2. Experimental process.** Rectangle boxes represent blocks, separated by lines representing single trials.

whole-body scanner, Sigma HDx, GE Healthcare, Milwaukee, Wisconsin, USA). In order to define the cTBS target area, we used MNI coordinates (x = 26, y = 258, z = 43) reported in Azañón et al. [23] (2010), who stimulated the right-hemisphere human homologue of macaque ventral intraparietal area. We therefore expected that cTBS would disrupt activity in the Superior Parietal Lobule (SPL; Area 7A) and Intraparietal Sulcus (IPS) of the right hemisphere. Stimulation locations were targeted using real-time neuronavigation software (TMS Neuronavigator, Brain Innovation, Maastricht, Netherlands).

In order to localize the stimulation area for each individual participant, the high-resolution scan was co-registered and normalized to the MNI template.

### Data processing and data reduction

All data processing was performed using customized functions scripted in Matlab 2018b (Mathworks, MA, USA). Centre-of-Pressure (CoP) data of the force plate was digitally low-pass filtered with a cut-off frequency of 10 Hz (dual-pass, 4th-order Butterworth). CoP position was differentiated to obtain CoP rate-of-change in m/s(dCoP). In order to characterize balance recovery, we followed a similar approach as applied in Johannsen et al. [4]. The standard deviation of the medio-lateral dCoP (SD dCoP) was calculated for each of 13 temporal bins of 1 s duration before and after the moment of the perturbation. A period of 3 s duration before the perturbation served as an intra-trial sway baseline. Across the 10 post-perturbation bins demonstrating stabilization, we fitted from an exponential decreasing non-linear regression $x(t) = C + A^* e^{\left(-\frac{t}{B}\right)}$, from which we obtained the function parameters A (intercept), B (time constant) and C (asymptote). The intercept is derived from the body sway at perturbation (t = 0) and therefore reflects the immediate effect of the perturbation. The time constant represents the rate of stabilization of body sway after the perturbation with shorter time constants

indicating faster stabilization. The third parameter, the asymptote, indicates the level of steady-state long-term stabilization.

EMG recordings were band-pass filtered between 10 and 500 Hz, rectified and smoothed by a moving average with 15ms width to obtain the EMG activity envelope of a muscle. For each muscle we extracted peak amplitude, indicating the amount of phasic activity directly following a perturbation and the area-under-the-curve of the activity envelope as an indication of the tonic activity across an entire trial serving as an indication of general muscle activation. EMG activity was then normalized to the first baseline block for NT and LT respectively and percentage of change from baseline was calculated.

## Statistical analysis

Data of the robotic device was checked for failures to deliver a forced push with an abrupt impact and immediate withdrawal of the end-effector. Trials in which the robotic arm only continuously shoved participants were excluded. Only successful force pushes were included in the data analysis. Overall there was a success rate of 87%.

Only trials with exponential fits of greater than 75% explained variance were included in the subsequent statistical analysis. In total, 15% of trials did not reach this threshold and were excluded from the statistical analysis. In order to identify possible non-responders to the cTBS stimulation we applied a k-means cluster analysis. K-means cluster analysis is a unsupervised learning algorithm that tries to cluster data based on their similarity, once the amount of desired clusters is defined. We defined 2 clusters (Responder vs. Non-responder) that we wanted data to be grouped into. Data for the intercept, time constant, asymptote, peak amplitude and area under the curve were pooled together and clustered in the two groups of either responders or non-responders. We identified two possible non-responders, leaving us with 11 participants for the statistical analysis. Prior to analysis data was log transformed to fit normal distribution. Parameters were then analysed statistically using a linear mixed model, with four repeated-measures factors (1) hand contact (Touch vs. No Touch), (2) stimulation session (cTBS vs. Sham), (3) Test (pre- vs. post-stimulation) and (4) force push (1% vs 4% vs 7%): (Variable~Stimulation_Session+Hand_Contact+Test+Force_Push+Stimulation_Session*Hand_Contact+Stimulation_Session*Test+Stimulation_Session*Force_Push+Hand_Contact*Test+LT*Force_Push+Test*Force_Push+Stimulation_Session*Hand_Contact*Test+Stimulation_Session*Test*Force_Push+Stimulation_Session*Hand_Contact*Force_Push+Hand_Contact*Test*Force_Push+Stimulation_Session*Hand_Contact*Test*Force_Push + (1|Subjects)) (Table 2). Fixed effects were "Hand_contact", "Stimulation_Session", "Test" and "Force_Push". Force push was treated as continuous, the others as factors. A post-hoc analysis was carried out to clarify the effects of stimulation session on muscle activity. A linear model with three repeated-measures factors (1) Test (pre- vs. post-stimulation), (2) hand contact (Touch vs. No Touch) and (3) force push (1% vs. 4% vs. 7%) was carried out for both stimulation sessions (sham and cTBS) respectively: (Variable~Test+Hand_Contact+Force_Push+Test*Hand_Contact+Test*Force_Push+Force_Push*Hand_Contact+Test*Hand_contact*Force_push + (1|Subjects)).

We also performed an analysis to investigate progression of sway over time with three repeated-measures factors (1) Block (progression over time), (2) hand contact (Touch vs. No Touch) and (3) stimulation session (cTBS vs. Sham): (Variable~Stimulation_Session+Hand_Contact+Block+Stimulation_Session*Hand_Contact+Stimulation_Session*Block+Stimulation_Session+Hand_Contact*Block+LT+Block+Stimulation_Session*Hand_Contact*Block+Stimulation_Session*Block+Stimulation_Session*Hand_Contact+Hand_Contact*Block+Stimulation_Session*Hand_Contact*Block + (1|Subjects)) (Table 3). We also

**Table 2. Results for Centre of Pressure and EMG.**

| Measure | P value | | | | | | | | |
|---|---|---|---|---|---|---|---|---|---|
| | Light Touch F (1,231) | Test F (1, 231) | Push Force F(2, 231) | Light Touch x Test F(1, 231) | Stimulation protocol x Test F(1, 231) | Test x Push Force F(1, 231) | Light Touch x Push Force F (2, 231) | Light Touch x Stimulation Protocol F(1, 231) | Stimulation protocol x Light Touch x Test F(1, 231) |
| | Centre of Pressure | | | | | | | | |
| Intercept | < .01 | < .001 | < .001 | < .05 | NS | NS | NS | NS | NS |
| Slope | NS | NS | < .05 | NS | NS | NS | NS | NS | NS |
| Constant | < .001 | < .001 | < .001 | < .001 | NS | NS | NS | NS | NS |
| | Tibialis Anterior | | | | | | | | |
| EMG Integral | < .001 | < .001 | NS | < .05 | < .001 | NS | NS | < .05 | NS |
| Peak Amplitude | < .001 | < .01 | < .01 | NS | NS | NS | NS | NS | NS |
| | Gastrocnemius | | | | | | | | |
| EMG Integral | NS | < .01 | NS | NS | NS | NS | NS | NS | NS |
| Peak Amplitude | < .001 | < .001 | NS | NS | < .05 | NS | NS | < .05 | NS |
| | Soleus | | | | | | | | |
| EMG Integral | NS | NS | NS | NS | NS | NS | NS | NS | NS |
| Peak Amplitude | < .05 | NS | < .001 | NS | NS | NS | NS | NS | NS |

performed a post-hoc analysis with specific focus on the first four blocks before the stimulation (Variable ~ Stimulation_Session + Hand_Contact + Block + Stimulation_Session*Hand_Contact + Stimulation_Session*Block + Hand_Contact*Block + Stimulation_Session*Hand_Contact* Block + (1 | Subjects)), investigating whether stimulation protocol had an influence in the pre-test already. This would hint at a session effect rather a stimulation effect.

For statistical significance, a p-value of 0.05 was used. Statistical analysis was carried out using the lme4 package in R-statistics (R version 3.4.0). Model estimates of the two main linear mixed models can be found in the supporting information.

## Results

### General sway analysis

Fig 3 shows illustrative data of one participant, averaged over all conditions. After the perturbation, the C7 body marker is deflected laterally accompanied by an excursion of the differentiated CoP signal. EMG activity of the Gastrocnemius rises to produce the required torque to compensate the perturbation. As a result, the CoP is accelerated into the opposite direction and C7 returns to the baseline position. EMG activity and CoP settle at pre-perturbation levels again until the end of the trial.

### CoP stabilization

Light Touch improved the immediate sway response to the perturbation compared no touch (Table 2). As can be seen in Fig 4, participants showed lower intercepts independently of the type of stimulation. Post hoc analysis revealed a significant effect of block, which is the progression over all 12 blocks (Table 3).

**Table 3. Results for analysis of gradual decrease.**

| Measure | | P value | | | | | | |
|---|---|---|---|---|---|---|---|---|
| | | Stimulation Protocol F(1,238) | Light Touch F (1,238) | Block F (1,238) | Stimulation Protocol x Light Touch F(1,238) | Stimulation protocol x Block F(1,238) | Light Touch x Block F(1,238) | Stimulation protocol x Light Touch x Block F(1,238) |
| | Centre of Pressure | | | | | | | |
| | Intercept | NS | < .001 | < .05 | NS | NS | NS | NS |
| | Slope | NS | NS | NS | NS | NS | NS | NS |
| | Constant | NS | < .001 | < .001 | NS | NS | NS | NS |
| | Tibialis Anterior | | | | | | | |
| | EMG Integral | < .001 | < .001 | < .001 | < .05 | < .001 | NS | NS |
| | Peak Amplitude | < .001 | < .001 | < .001 | < .05 | < .05 | NS | NS |
| | Gastrocnemius | | | | | | | |
| | EMG Integral | < .05 | NS | NS | NS | NS | NS | NS |
| | Peak Amplitude | < .01 | < .001 | < .001 | NS | NS | NS | NS |
| | Soleus | | | | | | | |
| | EMG Integral | NS | NS | NS | NS | NS | NS | NS |
| | Peak Amplitude | < .05 | < .05 | < .05 | NS | NS | NS | NS |

The effect can be derived from Fig 4 as well, showing a gradual decrease over time. Additionally, stronger lateral push forces resulted in higher intercepts (Fig 5A).

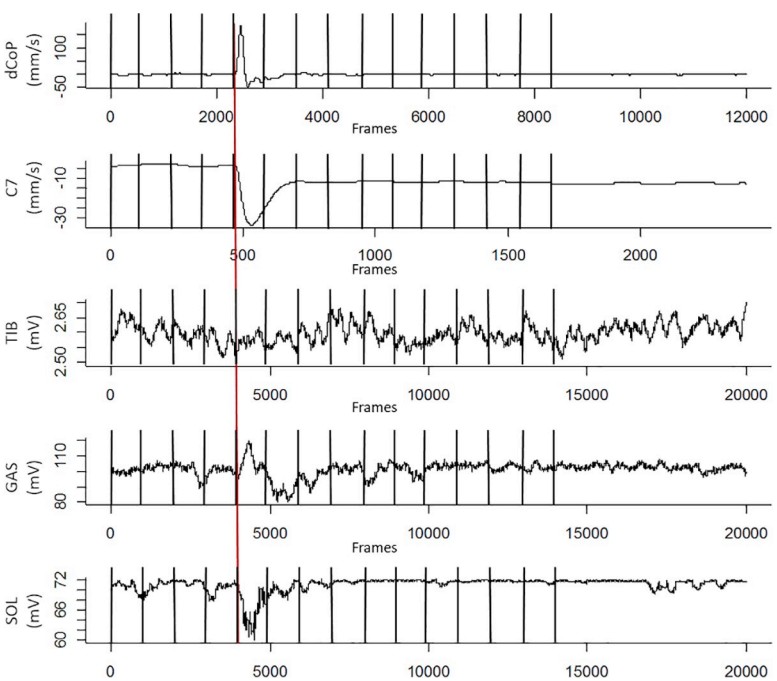

**Fig 3. Illustrative data of one participant averaged time course over all conditions of sway (ML dCoP (mm/s)), the C7 marker (mm/s), and the muscle response of the Tibialis Anterior (mV), Gastrocnemius (mV) and Soleus (mV).** The red line indicates the time of perturbation. Black vertical lines represent time bins of 1 second.

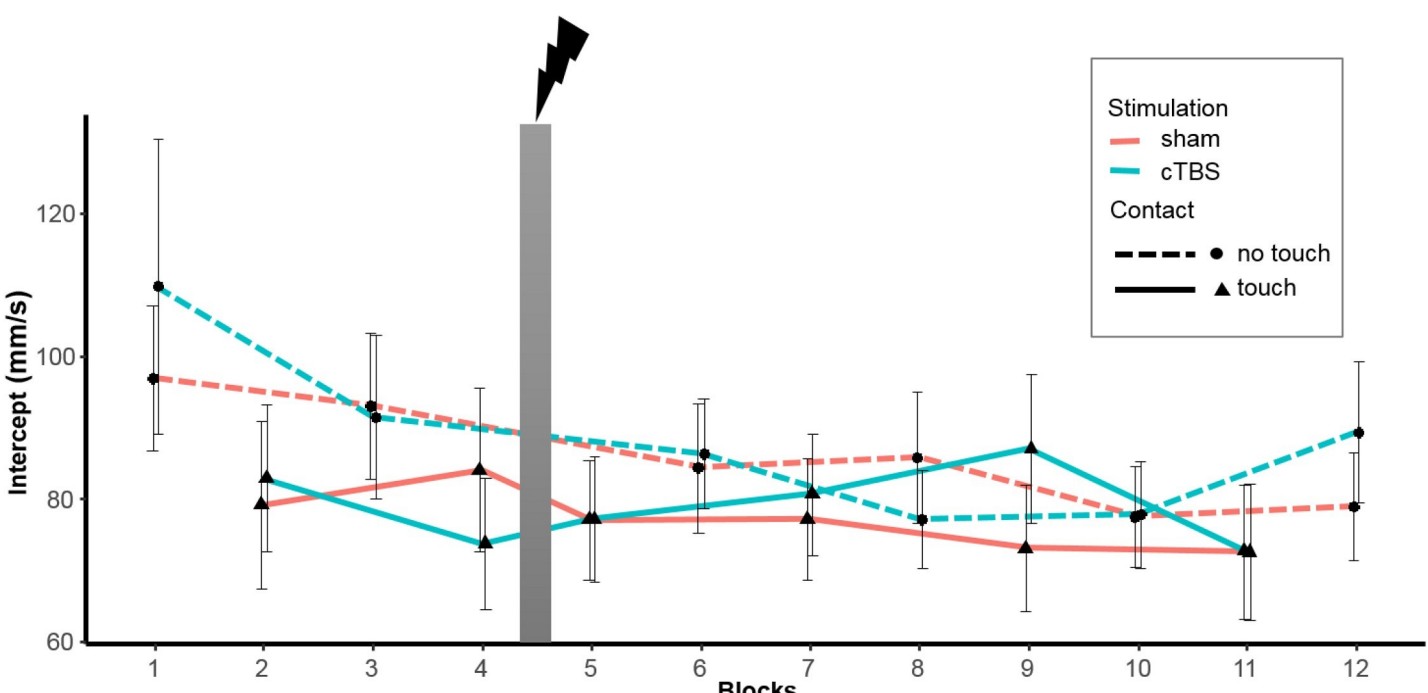

**Fig 4. Progression of averaged intercept of the body sway at perturbation as a function of contact condition (Touch/No Touch) and stimulation protocol (sham/cTBS).** Wide grey vertical line represents stimulation (Blocks left to it are pre-test, blocks right to it are post-test). Error bars indicate standard error.

The compensation time constant was only affected by push force. Similar to the immediate effect of the perturbation on sway, steady-state asymptote was reduced with Light Touch Independently of the type of stimulation (Table 2). Stronger pushing forces lead to a more variable postural steady state as indicated by higher asymptotes (Fig 5B). Asymptote showed a decrease of 15% in both the 1% and 7% force push condition and 20% decrease in the 4% force push

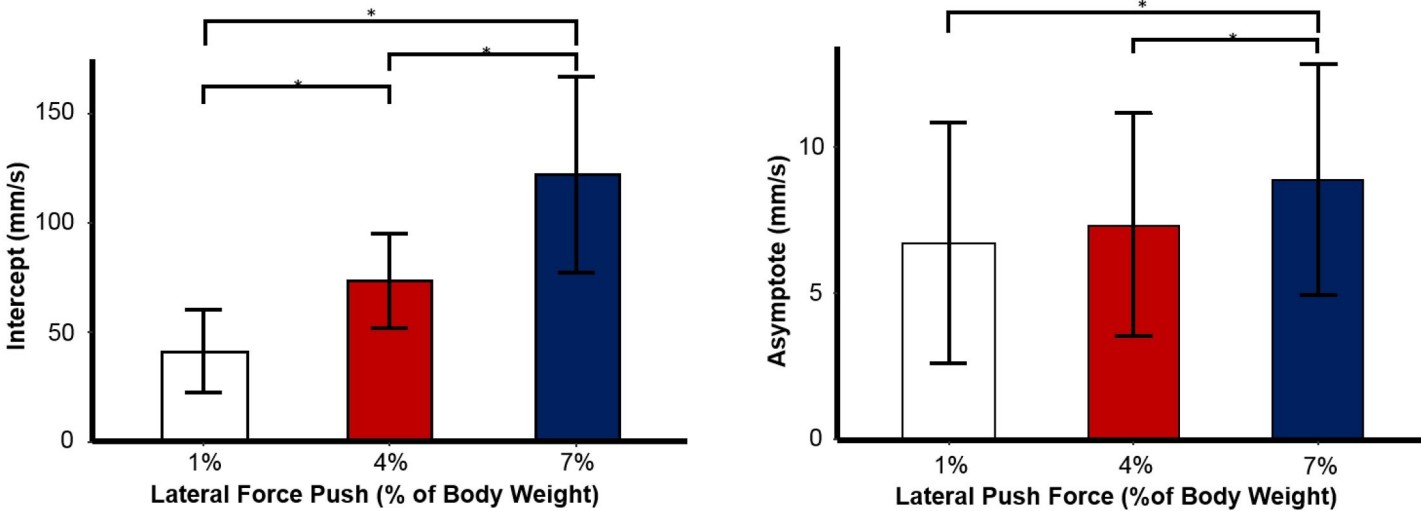

**Fig 5.** A) Averaged Intercept of the body sway at perturbation as a function of lateral push force (% of Body Weight). B) Averaged Asymptote of the body sway at perturbation as a function of lateral push force (% of Body Weight). Error bars indicate standard error.

compared to the pre-test. In addition, the asymptote also showed an interaction between Light Touch and intra-session testing (Table 2). We see the highest value during no touch in the pre-test. Asymptote values decrease in the post test even without Light Touch. However, we also see that with Light Touch asymptote values are already decreased in the pre-test. Even though with Light Touch asymptote values do not decrease further compared to the pre-test, there is a significant difference between post-test levels (p = .003), with smaller asymptote values when utilizing Light Touch (Fig 6). Post hoc analysis revealed again a gradual decrease over time, independently whether Light Touch was established or not (p < .001) (Table 3).

## EMG

Tibialis Anterior activity was affected by Light Touch and intra-session testing. Interactions between intra-session testing and stimulation protocol as well as between Light Touch and intra-session testing were found. General Tibialis Anterior activity decreased with the utilization of Light Touch. We saw that the highest level of general muscle activity (EMG integral) was expressed in the pre-test of the no touch condition, but decreased in the post-test. During the pre-test with Light Touch Tibialis Anterior activity already showed a lower level compared to no touch. Post hoc analysis of the two stimulation protocols revealed a significant effect of test (pre vs. post) for the Tibialis Anterior (p < .001) (Fig 7). Similar to the progression of sway we found gradual decrease of muscle activity over the progression of the 12 blocks (Figs 8 and 9). Post hoc test of the first four blocks before stimulation revealed no significant effect of stimulation session, showing that stimulation session is indeed an effect of the utilized stimulation rather than a general difference between sessions. Post hoc test did reveal a significant effect of Light Touch (p < .001) and Block (p < .05).

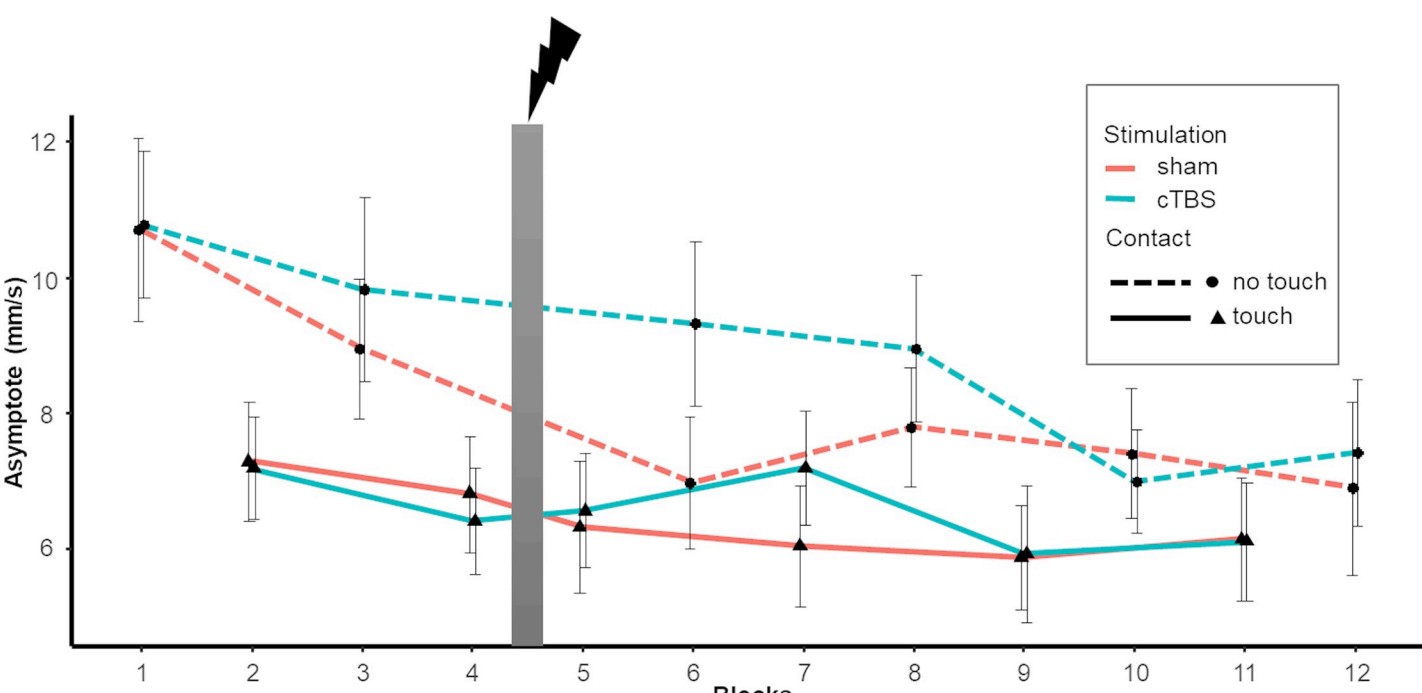

**Fig 6. Progression of averaged asymptote of the body sway at perturbation as a function of contact condition (Touch/No Touch) and stimulation protocol (sham/cTBS).** Wide grey vertical line represents stimulation (Blocks left to it are pre-test, blocks right to it are post-test). Error bars indicate standard error.

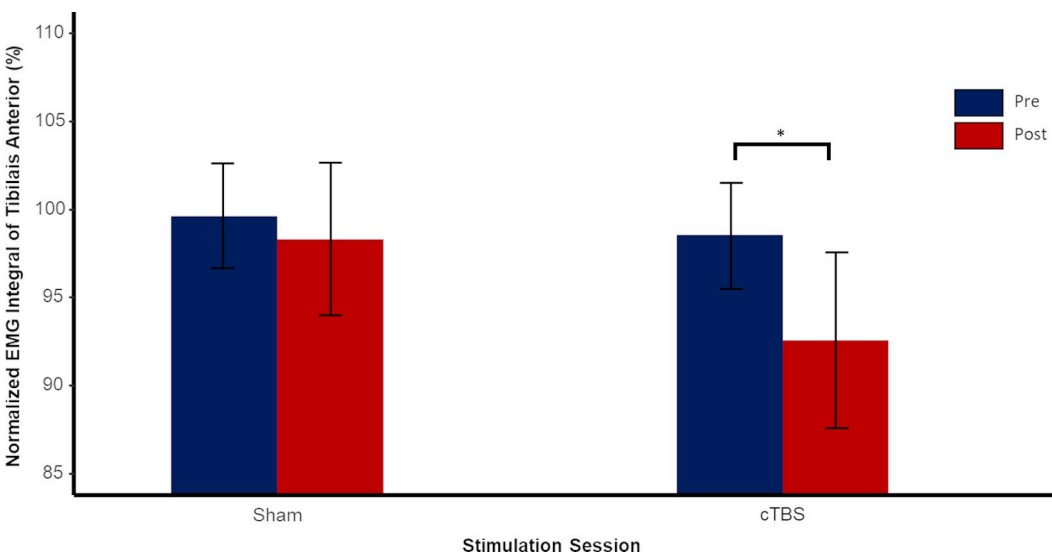

**Fig 7. Normalized EMG Integral of Tibialis Anterior as a function of Test (Pre/Post) and stimulation protocol (sham/cTBS).** Error bars indicate standard error.

Looking at the decrease in percentages, we see that in the 1% and 7% force push condition EMG integral decreases 13% and 11% respectively, while the 4% force push condition shows a greater decrease with 16%. Interestingly, cTBS stimulation showed greater decreased levels of muscle activity of the Tibialis compared to sham. Following sham stimulation muscle activity is decreased by 11% but after cTBS we saw a decrease of 16%. As can be derived from Table 3 post hoc analysis showed a significant interaction of stimulation protocol and intra-session testing.

In terms of peak amplitude of muscle activity directly following the perturbation, Gastrocnemius, Tibialis and Soleus all showed lower peak activity amplitudes with Light Touch

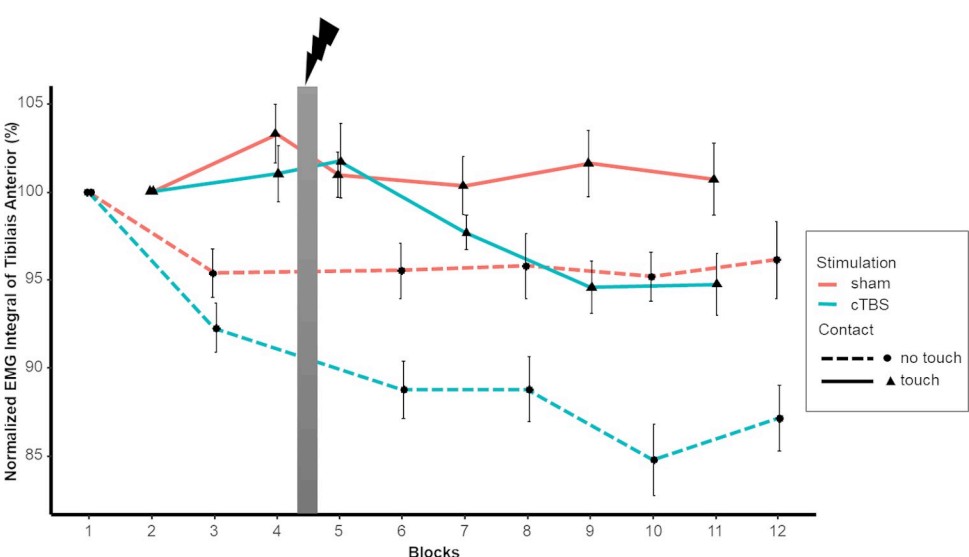

**Fig 8. Normalized EMG Integral of Tibialis Anterior as a function of contact condition (Touch/No Touch) and stimulation protocol (sham/cTBS).** Wide grey vertical line represents stimulation (Blocks left to it are pre-test, blocks right to it are post-test). Error bars indicate standard error.

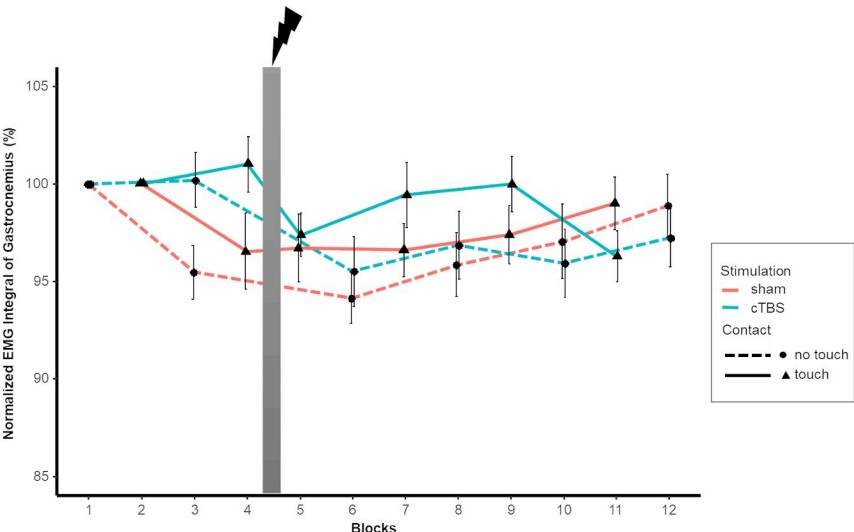

**Fig 9. Normalized EMG Integral of Gastrocnemius as a function of contact condition (Touch/No Touch) and stimulation protocol (sham/cTBS).** Wide grey vertical line represents stimulation (Blocks left to it are pre-test, blocks right to it are post-test). Error bars indicate standard error.

compared to No Touch (Table 2). Finally, a significant interaction between stimulation protocol and intra-session testing was observed for peak amplitude of the Gastrocnemius. Post-hoc analysis showed a differences between stimulations protocols. There was a significant effect of test for Gastrocnemius $p < .01$ for the cTBS stimulation, while after sham no effects were found. Similar to the stimulation effects of the EMG integral, we see a decrease of peak activity after cTBS stimulation, while it stays the same after sham.

## Discussion

Our study pursued two main objectives. The first was to investigate whether light fingertip contact improves balance compensation following a perturbation unpredictable in its relative force so that generation of a context-specific central postural set would be hindered. The second was to assess the role of the right posterior parietal cortex for the control of postural stiffness by disrupting the rPPC using continuous theta burst stimulation. We expected strong effects of light fingertip contact on body sway and muscle activations before, at and after a perturbation indicative of Light Touch feedback resulting in improved postural stability. Disruption of rPPC, on the other hand, was expected to hinder facilitation of sway stabilization with Light Touch but also affect the immediate response to a perturbation and sway stabilization by induced greater postural stiffness.

### Facilitation of body sway control with light touch

Baseline sway before a perturbation was reduced by Light touch in line with previous studies assessing steady-state postural sway [1]. At the perturbation, Light Touch reduced the immediate response as well as the asymptotic post-perturbation steady state. In addition, activity of the Tibialis Anterior and Gastrocnemius was reduced with Light Touch. Similar results were found when investigating Light Touch benefits on balance stabilization following a sudden backward perturbation [5,6]. Light Touch led to smaller amplitudes of CoP displacement and decreased muscle activity of the Gastrocnemius. Martinelli et al. [5] argued that usually large body oscillations are prevented primarily through torque production around the ankles and

that smaller displacement during Light Touch in return requires less muscle activation to produce smaller required correcting torque. Decreased general muscle activity (EMG Integral) in Tibialis Anterior across an entire perturbation trial agrees with this interpretation.

Against our expectations, Light Touch did not reduce the time constant of compensation following a perturbation. This observation contrasts with previous findings [4,5,6]. Johannsen and colleagues [4] observed shorter stabilization time constants with Light touch following both self-imposed as well as externally imposed perturbations. Similarly, Martinelli at al. [5] found reduced CoP sway during stabilization with Light Touch. However, their Light Touch effects for stabilization were limited to the most challenging conditions without vision while standing on a compliant surface. In all previous perturbation studies, that assessed the effect of augmented self-motion feedback with Light Touch, participants were tested in a normal bipedal stance posture with the perturbation in the antero-posterior direction [3,4,5,6]. In our present study, participants kept a tandem Romberg posture with a perturbation in the medio-lateral direction. Failed generalization of the Light Touch benefit to the time constant of balance stabilization in the context of the present study could indicate that the benefits of Light Touch for active stabilization could be highly context-specific. A central postural set represents the sensorimotor context of a postural task including the available sensory channels and current mechanical constraints [24]. Stance with Light Touch will also resemble a specific central postural set adjusted to the current task requirements such as the inclusion of a specific spatial frame of reference centred at the contacting finger or the trunk depending on the task [25,26]. If the postural context involves a balance perturbation, the task set will also represent the anticipated consequences of a known perturbation as well as any appropriate postural responses. For example, exposure to a sequence of horizontal support-surface perturbations with the same amplitude and velocity results in an appropriately scaled initial response of the agonist muscle, in contrast randomizing perturbations with respect to amplitude and velocity will result in a default response, partly determined by the strength the preceding perturbation [27]. In our current study, participants had to alternate between central postural sets with and without finger Light Touch in blocks of six trials each. Within each block the sequence of the perturbation forces was randomized and therefore unpredictable in its magnitude. The absence of any indications of Light Touch facilitation of dynamic stabilization in the current study implies a distinction between context-invariant or context-sensitive elements of a central postural set. Context-sensitive or rate-of-change-dependent components, such as an adequate compensation strategy following a perturbation, might have been excluded from the Light Touch central postural set or alternatively were impossible to implement due to the unpredictability of the experienced perturbations. It should be noted here that we did not find a direct influence of Light Touch in terms of shorter stabilization of the time constants. However, participants with a lower intercept but a constant time constant would reach their steady state sway earlier. In this regard, it might be possible that a strategy that even further decreases the time constant was deemed redundant, given that participants already reached their steady state faster.

Disruption of the rPPC did not interfere with the processing of fingertip haptic feedback for the stabilization of body sway following a perturbation. This confirms our previous study, where we showed that disruption of the rPPC did not affect the integration and utilization of Light Touch in a quiet stance context [17]. The present study generalizes this observation to more dynamic postural contexts involving external perturbations. This leaves us with a conundrum as the rPPC has been considered an important brain area that represents peri-personal space [28] and performs coordination transformation processes for mapping local tactile stimulation into hand-centered, head-centered, or trunk-centered spatial frames of reference [29,30]. Thus it seems likely that disruption of the rPPC does not alter the postural effects of

Light Touch sensory augmentation. As for the reason why, it is possible that a central postural set for the control of body sway with Light Touch makes use of more limb-cantered body representations without involvement of a predominantly spatial reference frame or egocentric representation. Dolgilevica and colleagues [31] proposed a conceptual framework which emphasizes the role of body representations such as the postural configuration of the body as well as the size and shape of body segments in the spatial localization of touch. In a previous study, we observed effector-specific differences between participants' dominant and non-dominant hand in terms of sway after-effects following sudden removal of a Light Touch reference [32]. The after-effect, that is the time to return to no touch baseline sway, was prolonged when the dominant hand was used to keep the Light Touch contact. As our participants were all right-handed, the observation implies that involvement of the left-hemisphere delayed switching between sets by keeping the Light Touch central postural set active for longer [32]. Thus, the control of body sway with Light Touch but without visual feedback may rely more on representations of somatotopy in the secondary somatosensory cortex [33] than representations of external space in the posterior parietal cortex.

## Control of postural stabilization following the perturbation

In our previous cTBS study involving a quiet stance situation, we found that disruption of the right PPC leads to a decrease of the general sway variability [17]. We attributed this reduction in sway to a disrupted process for the continuous exploration of the body's postural state [34] resulting in reduced inhibition of a process controlling postural stiffness [34]. Therefore, we expected that the postural perturbation paradigm of the present study would provide us with more direct evidence of an increase in postural stiffness following disruption of the rPPC. For example, reduced body sway in a steady postural state as well as a more rigid response to the lateral push, such as a reduced immediate effect of the perturbation on body sway but a prolonged time constant of stabilization, could be indicative of increased postural stiffness with reduced flexibility. The influence of postural stiffness on compensation of a balance perturbation has previously been shown by Horak and colleagues [35] testing Parkinson's patients, whose rigidity has been lowered by levodopa replacement therapy. Following support-surface translations these participants expressed less resistance and faster Centre-of-Mass displacement.

Jacobs and Horak [7] assumed that contextual cues of an impending perturbation are used to optimize anticipatory postural adjustments. Based on that assumption, Smith et al. [36] analysed the effects of support translations on anticipatory postural adjustments testing how different amplitudes of support surface translations in combination with different cuing conditions influences optimization of anticipatory postural adjustments. Displacement amplitude was either cued by means of repetitive, blocked perturbations, or a random sequences of displacement amplitudes of uncued perturbations was delivered. In the blocked sequences, CoP under the feet showed a slower initial displacement following perturbations as compared to the random sequences. The authors interpreted the result as supporting the notion that postural control is optimized when contextual cues are given prior to the perturbation. The exposure to similar perturbations across trials in a block, however, may have induced optimization of postural responses by adaptive motor control processes and not through contextual cues alone [36]. Coelho et al. [37] investigated whether optimized postural responses are a result of contextual cuing or whether they are dependent on motor experience. They were able to show that block sequence of perturbations leads to the generation of more stable automatic postural responses in comparison to the serial and random perturbation sequences. During block sequence perturbation lower body sway amplitude, decreased displacement velocity and

longer delays of activation onset of leg distal muscles were found. They interpreted these results as optimized postural responses in the block sequence due to adaptive processes underlying repetitive perturbations over trials rather than to processing of contextual cues [37]. To better understand how the postural control system adjusts postural responses following a specific type of perturbation, Kim et al. [38] exposed participants to forward trunk pushes of 5 different strengths in randomized order and estimated the gradual scaling of the sensory feedback gain. After comparing the observed feedback gain scaling to perturbations expressed following support surface translations [39], they concluded that the postural control system seems to select a feedback gain set according to the current postural context as characterised by the type of a perturbation and biomechanical constraints. Although Kim et al. [38] favoured a feedback gain interpretation, they could not exclude the possibility of situation-specific changes in dynamic parameters such as joint stiffness and damping.

In our present study we found results indicative of an adaptive process in terms of lower leg muscle activity and steady state sway, with a general decrease over time, independently whether Light Touch was used or not. This supports the idea that exposing people repetitively to a perturbation leads to an optimization of the postural response. Interestingly, this adaptive process was present although participants were perturbed to a randomized sequence of three different force pushes within one block. Given the range of the perturbations with a small, medium and strong force push, one possibility is that instead of finding three strategies against the perturbation force, the postural control systems settled for a compromise across the three forces and prepared for a medium configuration. If this were the case we would expect to see greater improvement, respectively greater decrease of muscle activity and postural sway in the medium force push condition. Looking at the decrease in percentages, this was the case. While in the small and strong force push condition we see a reduction in the EMG integral of the Tibialis of 13% and 11% respectively, the medium force push condition shows the highest decrease with 16%. Similar results can be found for the asymptote, with a decrease of 15% in both the small and strong force push condition and 20% decrease in the medium force push.

Unexpectedly, cTBS stimulation resulted in more decreased levels of activity of the Tibialis anterior and peak activity of the Gastrocnemius compared to sham stimulation. This observation contrasts with tonic activity of the Gastrocnemius, where activity stayed relatively the same over time, independently of the type of stimulation. Sozzi and colleagues [40] investigated the individual role of the lower leg muscles during standing in tandem Romberg stance and reported roles of the muscles specific to individual balancing functions. They concluded that while the soleus supports the body against gravity, the Tibialis Anterior and the peroneus stabilize the body in the medio-lateral direction. This supports our conclusion that the greater reduction in Tibialis anterior activity is tied to an improved postural adaptation following cTBS of the rPPC.

The decrease of muscle activity in the Tibialis Anterior should not be mistaken as a direct influence of the rPPC disruption on muscle activity, but rather as a result of a centrally mediated adaptation of postural control to the challenges of a perturbation. If we assume that reduced lower leg muscle activity indicates an experience-dependent optimization of the postural adjustments, then we can conclude that rPPC disruption enhanced anticipation of the disturbing effects of the perturbation. In Kaulmann et al. [17], we argued that rPPC may be involved in a process with generates postural sway to actively explore the postural stability state, which might normally interact with a postural stiffness control process in a reciprocal inhibitory manner. Thus, cancellation or disruption of a process represented in the rPPC for exploring the postural state might lead to a clearer feedback-dependent signal used for the prediction of the effects of an externally imposed external perturbation and the optimization of any compensatory responses.

There is ample evidence, however, that points to the role of brain areas other than the cerebral cortex in the adjustment of postural responses to external perturbations of balance. For example, Thach and Bastian [41] reported that the cerebellum is involved in the adaptation of response magnitude, as well as in the tuning of the coordination of postural responses based on practice and knowledge. This was in line with Horak and Diener [42], who demonstrated that patients with cerebellar lesions are unable to scale the magnitude of their postural responses to predicable amplitudes of surface translations. Also involvement of the basal ganglia in postural responses following external perturbations as illustrated by Parkinson's disease resulting in the inability to modify postural responses to a perturbation [43]. For example, healthy subjects are able to change postural synergies immediately after a single exposure, while individuals with Parkinson's disease require several trials to adjust their responses [44]. Thus, we do not claim that the rPPC is exclusively involved in the adaptation to a postural perturbation but that the region nevertheless resembles an important component of a network of brain regions controlling postural stiffness and adaptation.

## Limitations

We have no direct indicator of the neural effect induced by cTBS stimulation at the target cortical area. Therefore, we cannot assume without reservation that cTBS did indeed cause local inhibition of the rPPC as the region, being primarily involved in sensorimotor integration for movement control, does not project directly to end-effector specific areas in the primary motor cortex that could have validated its effectiveness. Therefore, the evidence presented by our study for a role of the rPPC in the adaptation of postural responses to unpredictable perturbations must be considered as circumstantial only. A subsequent study needs to follow-up our observations by being more properly designed to evaluate sensorimotor learning of the perturbations and which validates the disruption of rPPC by cTBS using a different probe task, for example assessing visual attention.

## Conclusion

We found a strong effect of Light Touch, which resulted in improved stability following an unpredictable perturbation. Light Touch decreased the immediate sway response, as well as the steady state sway following re-stabilization. Decreased sway is accompanied by reduced muscle activity of the ankle Tibilais Anterior. We assume that the improved sway response lead to increased stability, which required less torque production around the ankles in order to stabilize the body. However, we did not find an improvement of the time constant in response to the perturbation with Light Touch. This contrasts with studies that investigated the benefit of Light Touch when compensating a perturbation in the sagittal plane, while standing in normal bipedal stance. The lack of improvement might be a result of a different postural context or the unpredictability of the force of the perturbations. We observed a gradual decrease of muscle activity, which is indicative of an adaptive process in terms of lower leg muscle activity, following exposure to repetitive trials of perturbations. This supports the idea that exposing people repetitively to a perturbation leads to an optimization of the postural response. Given the range of the perturbations we suspect that the postural control system settled for a compromise across the three different perturbation forces and prepared for a medium configuration. This is supported by the notion that we see greater decrease of muscle activity in the medium force push condition. Regarding the effects of the disruption of the rPPC we were not able to confirm our hypothesis that disruption of the rPPC leads to increased postural stiffness. However, we did find an unexpected effect of cTBS stimulation in terms of improvements of the aforementioned adaptive process. After disruption of the rPPC muscle activity of the Tibialis

Anterior is decreased even greater, compared to sham. From that we can conclude that rPPC disruption enhanced the intra-session adaptation to the disturbing effects of the perturbation.

## Supporting information

**S1 File.**
(DOCX)

## Acknowledgments

We like to thank Dr. Thomas Stephan for his help with collecting the anatomical brain scans.

## Author Contributions

**Conceptualization:** David Kaulmann, Leif Johannsen.

**Formal analysis:** David Kaulmann.

**Investigation:** David Kaulmann, Matteo Saveriano.

**Resources:** Dongheui Lee, Joachim Hermsdörfer.

**Supervision:** Joachim Hermsdörfer, Leif Johannsen.

**Writing – original draft:** David Kaulmann.

**Writing – review & editing:** Leif Johannsen.

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
