## [Decision Letter · Decision Letter 0]

20 Jan 2020

PONE-D-19-35562

Stabilization of body balance with light touch following a mechanical perturbation: Adaption of sway and disruption of right Posterior Parietal Cortex by cTBS

PLOS ONE

Dear Mr Kaulmann,

Thank you for submitting your manuscript to PLOS ONE. After careful consideration, we feel that it has merit but does not fully meet PLOS ONE’s publication criteria as it currently stands. Therefore, we invite you to submit a revised version of the manuscript that addresses the points raised during the review process.

You will see that the reviewers voiced several major concerns, among others regarding data analysis, statistics and the conclusions drawn based on the data.

We would appreciate receiving your substantially revised manuscript by Mar 05 2020 11:59PM. To enhance the reproducibility of your results, we recommend that if applicable you deposit your laboratory protocols in protocols.io, where a protocol can be assigned its own identifier (DOI) such that it can be cited independently in the future. For instructions see: http://journals.plos.org/plosone/s/submission-guidelines#loc-laboratory-protocols

We look forward to receiving your revised manuscript.

Kind regards,

Andreas Kramer

Academic Editor

PLOS ONE

Journal Requirements:

2. Please provide further detail regarding your recruitment procedure and date.

3. Please revise the first sentence of your abstract.

4. We note you have included a table to which you do not refer in the text of your manuscript. Please ensure that you refer to Table 2 in your text; if accepted, production will need this reference to link the reader to the Table.

Reviewers' comments:

Reviewer's Responses to Questions

**Comments to the Author**

1. Is the manuscript technically sound, and do the data support the conclusions?

Reviewer #1: Partly

Reviewer #2: Partly

2. Has the statistical analysis been performed appropriately and rigorously? 

Reviewer #1: No

Reviewer #2: Yes

3. Have the authors made all data underlying the findings in their manuscript fully available?

Reviewer #1: Yes

Reviewer #2: Yes

4. Is the manuscript presented in an intelligible fashion and written in standard English?

Reviewer #1: Yes

Reviewer #2: Yes

5. Review Comments to the Author

Reviewer #1: Comments

The authors are interested in assessing the role of the rPPC in regard to postural control with and without light touch during postural perturbation. To test the involvement of the rPPC, the authors used a cTBS protocol to disrupt the cortical region. My comments here are mostly limited to the methodological parts of this manuscript since I am not well aware of the literature related to light touch or postural control. Although this study is interesting, many important methodological parts are not described, which prevent a correct interpretation of the results. Several limitations issues are required to be included and discussed in the manuscript. Statistics must be better described and possibly re-done (depending on the error structure of the models). The interpretation in regard to the role of the rPPC in stabilisation and adaptations are not really supported by the present data.

Methods

-The task is correctly described but unless I missed it, there was no description of the instructions given to the subject. This is important as it can obviously affect their behaviour following the robotic push (e.g. “resist the push” vs. “stand still” vs. “limit your centre of mass movements”).

- I praise the authors for controlling stimulation site with both neuronavigation and anatomical support. However, the description of the protocol needs to be more detailed. cTBS intensity relies on the correct assessment of a motor threshold (active or passive depending on the literature). Please explain how the motor threshold was obtained, and in which muscle. Please comment on the fact that the MT was assessed in a place different than the stimulation site (if I understand correctly). How the authors can be certain that the MT would be the same at different cortical position (given the possible differences in axons orientation due to possibly different cortical folding)? Moreover, also possibly due to a difference in cells orientation but also in activity, the region stimulated may not respond to cTBS the same way than the FDI motor region. See for example https://doi.org/10.1016/j.clinph.2006.08.008. Further, you have not tested whether the cTBS stimulation was successful at disrupting the area. This is a big concern since it has been shown that cTBS has extremely variable effects (10.1016/j.clinph.2017.08.023). Therefore, in the present experiment, we do not know whether cTBS was able to disrupt the area, or if it did it whether it increased or decreased the region excitability. Please address these concerns in a limitation section and change your discussion accordingly. What was the procedure for the sham stimulation?

-Statistics. I salute the authors for using LMM. However, the statistical part is not detailed enough to understand how statistics were performed and how the readers should interpret results. What are the distributions of your data sets? If they are not normal, please at least indicate it to the reader (in the methods and in a limitation section), and data transformation or use of GLM is advised. Please details the models used: Are the 4 fixed effects implemented in the same model or in different models? How are the models hierarchized and the data clustered? What are the different levels within the factors (are they treated as factor or as continuous variable)? What are the random effects? In the statistic section it reads that there was no interaction between factors in the model, but then in the results it seems that there were interactions. Furthermore, it seems that the model was not fully “maximal” according to errors, which make it more prone to type I error (see Barr 2013). Please maximize the error structure including at the interaction level. How did you obtain p-values with the LMM, have you used the LmerTest package? You have performed post-hoc analysis. Please describe them. Given the model you used, these post hoc tests may not be necessary when looking directly at the models parameters. In the Tables, in the title line, there is a F(). What does it stand for? F-values? Adding parameters values (intercepts and betas) and their CI or errors would help the comprehension of the tables.

-In the Tables, what is the factor Test?

-What is the rationale behind separating analysis for peak EMG activity and EMG activity. How should we interpret a change in Peak activity but no change in general activity? This separation increases the number of tests performed, which in a frequentist paradigm increases the risks of false alarm (a change in peak but not in general activity and vice-versa seems actually to be more related to a false alarm than a solid physiological result).

Discussion

-In page 19, the authors talk about the lack of effect from the rPPC disruption and how it creates a conundrum with the existing literature. The physiological interpretations should be toned down since there was here no controls of the disruption effectiveness of the cTBS (associated with a low sample size). The authors should also discuss the fact that since they have not controlled for the effectiveness of the cTBS, the absence of effect is in no way conclusive in regard to the role of the rPPC in regard to posture stabilisation with/without light touch. Using the changes in EMG as a control of the effect of cTBS would not be very credible as well (see comments below).

-The authors states that cTBS may have reduced more Muscle activity after perturbation. However, if I read correctly the figure 7 (for the tibialis), the most extreme difference of EMG activity between the 2 sessions occurs before the cTBS stimulation (unless there is a problem in the legends?). I cannot assess whether this observation is similar for the peak activity of GAS since there is no related figure. Therefore, it seems probable to me that the difference observed is more due to variability than from an effect of cTBS. This point is supported by the fact that you obtained a significant interaction between blocks and stimulation only for general activity of for the peak activity but never for both at the same time (while we would expect it to happen, unless I am mistaken about the interpretation of such parameters). In any cases, the observed difference of 5% between sessions seems meaningless compared to the overall observed variability. Please comment this point and change the discussion/abstract/conclusion accordingly.

-In the abstract and in the conclusion the authors indicate that cTBS affected gastrocnemius activity, but in the discussion above it is said that cTBS changed Tibialis activity (p23) or all muscle activity according to the Tables. Maybe some precision should be added there?

-In the figure 7, the dashed or solid lines are not following what the legend says.

Reviewer #2: The presented study investigates 1) the effect of light touch on spontaneous sway amplitudes and sway responses following a push perturbation and 2) the effect of a disruption of the right hemispheric Posterior Parietal Cortices. I do not have an expertise in cTBS or similar methods. Therefore, the review does not address any cTBS related methodological aspects. This also greatly hampers my ability to judge, whether the rationale that rPPC should be involved in the tasks used in this study. Thus, my comments relate to the tasks and the study in general. Overall, the study appears to be reasonably well founded and conducted. However, there are some major aspects that should be addressed. Please find my concerns and comments below.

Major comments

1) The data-analysis process appears quite arbitrary and complicated while there is no obvious reason for some steps of this approach. Specifically, the process of binning and then fitting, as well as the reason for excluding bad fits should be – at least - clarified and justified.

2) Fig. 7: I guess that colors and line style are wrong. Otherwise the results would make absolutely no sense.

3) Conclusions: “However, we did found an unexpected effect of cTBS stimulation in terms of improvements of the aforementioned adaptive process. After disruption of the rPPC muscle activity of the Gastrocnemius is decreased even greater, compared to sham. From that we can conclude that rPPC disruption enhanced the intra-session adaptation of the disturbing effects of the perturbation.”

Maybe I missed something, but I am not convinced that this statement is supported by the data.

Minor comments

General comment: there appear to be several citation styles and inconsistencies in the manuscript. E.g. in line 67 „Dickstein and colleagues (2003) [3]“. So the references should be cleaned up throughout the manuscript.

Figure numbers are not correct.

In Figure 2 (as indicated on the figure itself) also the push conditions could be displayed. Also adding the scheme in Figure 1 into this figure might be possible and would the number of figures.

Line 34: The sentence appears to have some words missing „has been improve...“

Line 37-40: Are these two hypotheses related to each other? If yes, please clarify. If not, I would suggest to formulate them as two hypotheses and not connect them using ‚but‘.

Line 49: Here you state that „We were not able to confirm our hypothesis that disruption of the rPPC leads to increased postural stiffness.“ However, above you state that: „[…] sway stabilization would be generally impaired following disruption of the right Posterior Parietal Cortex.“

Maybe its better to state the same thing in the hypothesis formulation and in the results.

Line 136: The formulation of the hypotheses are a little cryptic. For example: what is meant by “relative benefit [… is] amplified”? Or the “[reduced] facilitation of sway stabilization”. Something closer to the parameters (e.g. reduced sway response) would be much more specific and easier to understand.

Line 160: referent should be references

Table 1: I don’t believe this table is necessary. If its left in the manuscript a more precise tiltle is required (“… for one example subject”)

line 217: “CoP position was differentiated to obtain CoP rate-of-change in seconds (dCoP).” Differentiating a position gives velocity – m/s and not s.

line 219: “before and after at the” delete at

Figure 3: x-axis labels are missing and x-axis should be given in seconds.

Line numbers are missing starting at page 14.

Page 14 bottom: weather should be whether

Fig. 6: legend for different colors are missing.

Fig. 7 and 8: I suggest to rename “general muscle activity” by something like “EMG integral”, since the current label is very confusing.

Page 17: the introduction of the discussion is much better in clarifying the hypotheses as compared to Abstract and Introduction. Maybe the latter could be reformulated along those lines.

Page 22: “cuing or hether they” is this supposed to mean whether?

Page 23: “asymptote, with a decrease of 15% in both the 1% and 4% force push condition and 20%” should be “1% and 7%”

Page 31: “However, we did found an unexpected effect of cTBS stimulation in terms of improvements of the aforementioned adaptive process.” Should be “we did find”.

6. PLOS authors have the option to publish the peer review history of their article (what does this mean?). If published, this will include your full peer review and any attached files.

Reviewer #1: No

Reviewer #2: Yes: Lorenz Assländer

---

## [Author Response · Author response to Decision Letter 0]

25 Feb 2020

Dear Editor,

we would like to thank the reviewers for their very valuable criticisms and comments regarding our study “Stabilization of body balance with light touch following a mechanical perturbation: Adaption of sway and disruption of right Posterior Parietal Cortex by cTBS”. We followed the reviewers’ advice and revised our manuscript thoroughly. Below we explain the changes made to the manuscript point-by-point.

Reviewer 1

1. The task is correctly described but unless I missed it, there was no description of the instructions given to the subject. This is important as it can obviously affect their behaviour following the robotic push (e.g. “resist the push” vs. “stand still” vs. “limit your centre of mass movements”).

We added information regarding the instructions given to the subject. Subject were merely instructed to stand as natural as possible, in order to avoid making the task of maintaining balance explicit (Line 184).

2. I praise the authors for controlling stimulation site with both neuronavigation and anatomical support. However, the description of the protocol needs to be more detailed. cTBS intensity relies on the correct assessment of a motor threshold (active or passive depending on the literature). Please explain how the motor threshold was obtained, and in which muscle. Please comment on the fact that the MT was assessed in a place different than the stimulation site (if I understand correctly). How the authors can be certain that the MT would be the same at different cortical position (given the possible differences in axons orientation due to possibly different cortical folding)? Moreover, also possibly due to a difference in cells orientation but also in activity, the region stimulated may not respond to cTBS the same way than the FDI motor region. See for example https://doi.org/10.1016/j.clinph.2006.08.008. Further, you have not tested whether the cTBS stimulation was successful at disrupting the area. This is a big concern since it has been shown that cTBS has extremely variable effects (10.1016/j.clinph.2017.08.023). Therefore, in the present experiment, we do not know whether cTBS was able to disrupt the area, or if it did it whether it increased or decreased the region excitability. Please address these concerns in a limitation section and change your discussion accordingly. What was the procedure for the sham stimulation?

We added additional information about the procedure of assessing the motor threshold (Line 237). We also added from which muscle it was measured. Given we implemented and followed standard procedure, we are positive that MT was determined correctly. The procedure served to individually standardize the stimulation strength since important factors such as thickness of the scull bone vary inter-individually more than across different cortical areas of an individual person. Stimulation of non-motor cortical areas with a fixed strength determined by M1 stimulation is a frequently used method (Siebner & Ziemann, 2007). We are however aware, that like with all such TMS protocols applied outside primary motor regions there is a degree of uncertainty regarding the effectives. Observed effects of stimulation seem likely to stem from inhibition of the cortical target area, given that the sham coil is not able to produce and thus induce cortical changes. Behavioral changes observed seem thus likely to be the result of disruption of the rPPC. We added information regarding the sham coil as well (Line 247). However, as the reviewer suggested, we added a limitation section pointing out that we were not able to assess whether excitability of the target area was indeed reduced. It is fair to point out, that due to the nature of the rPPC a direct assessment, which does not include fMRI or EEG, of a change of excitability is simply not possible. 

3. Statistics. I salute the authors for using LMM. However, the statistical part is not detailed enough to understand how statistics were performed and how the readers should interpret results. What are the distributions of your data sets? If they are not normal, please at least indicate it to the reader (in the methods and in a limitation section), and data transformation or use of GLM is advised. Please details the models used: Are the 4 fixed effects implemented in the same model or in different models? How are the models hierarchized and the data clustered? What are the different levels within the factors (are they treated as factor or as continuous variable)? What are the random effects? In the statistic section it reads that there was no interaction between factors in the model, but then in the results it seems that there were interactions. Furthermore, it seems that the model was not fully “maximal” according to errors, which make it more prone to type I error (see Barr 2013). Please maximize the error structure including at the interaction level. How did you obtain p-values with the LMM, have you used the LmerTest package? You have performed post-hoc analysis. Please describe them. Given the model you used, these post hoc tests may not be necessary when looking directly at the models parameters. In the Tables, in the title line, there is a F(). What does it stand for? F-values? Adding parameters values (intercepts and betas) and their CI or errors would help the comprehension of the tables.

We followed the advice of the reviewer and substantially revised our statistics section. In addition we firstly introduced a cluster analysis to be certain to identify possible non-responder to the stimulation. Data was checked for normal distribution. Since it did not meet normal distribution it was then log-transformed and LMM was applied again. Tables in the result section and the method section have been corrected accordingly. We also provided more detailed information regarding the used models. We also improved the model and maximized it. Used R-package for data analysis has also been added as information in the method section (Line 301 to 325).

4. In the Tables, what is the factor Test?

Factor Test stands for the evaluation of pre- vs. post-stimulation. Method section was corrected accordingly. In the Methods section this factor was previously named “effect of stimulation”, this has been corrected and “Test” is now used overall (Line 314).

5. What is the rationale behind separating analysis for peak EMG activity and EMG activity. How should we interpret a change in Peak activity but no change in general activity? This separation increases the number of tests performed, which in a frequentist paradigm increases the risks of false alarm (a change in peak but not in general activity and vice-versa seems actually to be more related to a false alarm than a solid physiological result).

General EMG activity and peak EMG activity describes two different factors for stabilization of balance. The peak EMG activity is the highest peak after the perturbation, usually indicative of the immediate perturbation response. The Integral of the EMG (general muscle activity) describes the tonic activation of the muscle throughout the whole trial (Line 291 – 294). We therefore considered general EMG activity an peak activity as independent measures. 

6. In page 19, the authors talk about the lack of effect from the rPPC disruption and how it creates a conundrum with the existing literature. The physiological interpretations should be toned down since there was here no controls of the disruption effectiveness of the cTBS (associated with a low sample size). The authors should also discuss the fact that since they have not controlled for the effectiveness of the cTBS, the absence of effect is in no way conclusive in regard to the role of the rPPC in regard to posture stabilisation with/without light touch. Using the changes in EMG as a control of the effect of cTBS would not be very credible as well (see comments below).

We added a limitation section to the discussion, discussing the possibility that cTBS did not alter cortical activity of the target area (see point 2) (Line 626 – 646).

7. The authors states that cTBS may have reduced more Muscle activity after perturbation. However, if I read correctly the figure 7 (for the tibialis), the most extreme difference of EMG activity between the 2 sessions occurs before the cTBS stimulation (unless there is a problem in the legends?). I cannot assess whether this observation is similar for the peak activity of GAS since there is no related figure. Therefore, it seems probable to me that the difference observed is more due to variability than from an effect of cTBS. This point is supported by the fact that you obtained a significant interaction between blocks and stimulation only for general activity of for the peak activity but never for both at the same time (while we would expect it to happen, unless I am mistaken about the interpretation of such parameters). In any cases, the observed difference of 5% between sessions seems meaningless compared to the overall observed variability. Please comment this point and change the discussion/abstract/conclusion accordingly.

First of all, we like to mention that due to the log transformation and the maximization of the model statistical outcomes have changed slightly. We now see a stimulation effect in both peak and general EMG activity. It is correct that it seems like the effect stems from the difference between the 2 sessions before cTBS stimulation. It is also correct, that there is a difference between the pre-tests of the two stimulation session. However, this general difference and activation amplitude can result from the placement of the EMG-sensor, which might have deviated slightly in these two sessions. Therefore, it is important to look at the stimulation effect in the two sessions separately. In order to clarify this effect, we carried out another post hoc analysis of the two session separated from another, looking at the mere pre vs. post stimulation effect. We see that there is no significant change of muscle activity following sham stimulation, while we observe a significant change after cTBS stimulation. Since the sham coil is not able to change cortical activity, it seems likely that the observed changes during the cTBS session are in fact a result of altered cortical activity of the rPPC. In order to visualize this better, we changed figure 7 and figure 8, displaying the change of percentage from the first block to the last.

8. In the abstract and in the conclusion the authors indicate that cTBS affected gastrocnemius activity, but in the discussion above it is said that cTBS changed Tibialis activity (p23) or all muscle activity according to the Tables. Maybe some precision should be added there?

We realized a mistake in the conclusion as pointed out by the reviewer. We corrected the conclusion accordingly (Line 664), the description in the discussion and in the tables was correct. 

9. In the figure 7, the dashed or solid lines are not following what the legend says.

Figure 7 has been modified. It now displays the percentages of change from the first block to the last as a function of contact condition (Light Touch vs. No Touch) and stimulation protocol (sham vs. cTBS). The same was done for Fig. 6. We believe this kind of display is a more fitting representation of the gradual decrease over time. 

Reviewer 2

1. The data-analysis process appears quite arbitrary and complicated while there is no obvious reason for some steps of this approach. Specifically, the process of binning and then fitting, as well as the reason for excluding bad fits should be – at least - clarified and justified.

Justification for data analysis process has been added to the method section (Line 275). The same procedure was used in a previous paper by Johannsen et al. 2007. We applied the same procedure in order to make results easier to compare, since the previous study also investigated the effects of light touch for the control of balance following perturbations. 

2. Fig. 7: I guess that colors and line style are wrong. Otherwise the results would make absolutely no sense.

Fig. 7 has been corrected. However, it now displays the percentages of change from the first block to the last as a function of contact condition (Light Touch vs. No Touch) and stimulation protocol (sham vs. cTBS). The same was done for Fig. 6. We believe this kind of display is a more fitting representation of the gradual decrease over time and the stimulation effect. 

3. Conclusions: “However, we did found an unexpected effect of cTBS stimulation in terms of improvements of the aforementioned adaptive process. After disruption of the rPPC muscle activity of the Gastrocnemius is decreased even greater, compared to sham. From that we can conclude that rPPC disruption enhanced the intra-session adaptation of the disturbing effects of the perturbation.”

Maybe I missed something, but I am not convinced that this statement is supported by the data.

In response to an advice of the first reviewer we introduced log transformation of the data and the maximization of the model (Line 311), statistical outcomes have changed slightly (See Result Tables). We now see a stimulation effect in both peak and general EMG activity, giving more support to our statement. Furthermore, we added another post hoc analysis of the two session separated from another, looking at the mere pre vs. post stimulation effect. We see that there is no significant change of muscle activity following sham stimulation, while we observe a significant change after cTBS stimulation.

4. General comment: there appear to be several citation styles and inconsistencies in the manuscript. E.g. in line 67 „Dickstein and colleagues (2003) [3]“. So the references should be cleaned up throughout the manuscript.

References have been cleaned up throughout the manuscript.

5. Figure numbers are not correct.

In Figure 2 (as indicated on the figure itself) also the push conditions could be displayed. Also adding the scheme in Figure 1 into this figure might be possible and would the number of figures.

Figure numbers have been corrected. Push condition cannot be displayed in Figure 2, since forces were always randomized and we wanted to avoid making the impression forces push conditions were the same in every trial of every participant.

6. Line 34: The sentence appears to have some words missing „has been improve...“

Sentence has been corrected.

7. Line 37-40: Are these two hypotheses related to each other? If yes, please clarify. If not, I would suggest to formulate them as two hypotheses and not connect them using ‚but‘.

Hypotheses have been clarified as two hypotheses and abstract corrected accordingly (Line 39 – 43) It now reads: “We hypothesized that the benefit of light touch would be amplified in the more dynamic context of an external perturbation, reducing body sway and muscle activations before, at and after a perturbation. Furthermore we expected sway stabilization would be impaired following disruption of the right Posterior Parietal Cortex as a result of increased postural stiffness.”

8. Line 49: Here you state that „We were not able to confirm our hypothesis that disruption of the rPPC leads to increased postural stiffness.“ However, above you state that: „[…] sway stabilization would be generally impaired following disruption of the right Posterior Parietal Cortex.“

Maybe its better to state the same thing in the hypothesis formulation and in the results.

We thank the reviewer for pointing out this inconsistency. We corrected this inconsistency in the hypothesis formulation as stated in the previous point.

9. Line 136: The formulation of the hypotheses are a little cryptic. For example: what is meant by “relative benefit [… is] amplified”? Or the “[reduced] facilitation of sway stabilization”. Something closer to the parameters (e.g. reduced sway response) would be much more specific and easier to understand.

Formulation of hypothesis has been clarified and is more consistent with the hypothesis formulation in the abstract. It now reads: “We hypothesized that the benefit of light touch would be amplified in the more dynamic context of an external perturbation to balance, improving the compensation response. We also expected that the immediate response to a perturbation and sway stabilization in terms of its time constant would be affected expressing an increase in postural stiffness following rPPC disruption.” (Line 161 - 166)

10. Line 160: referent should be references

Word has been corrected.

11. Table 1: I don’t believe this table is necessary. If its left in the manuscript a more precise tiltle is required (“… for one example subject”)

We left the table in the manuscript. We believe it is important for readers to know what force the participants were exposed to on average. We did formulate a more precise title.

12. line 217: “CoP position was differentiated to obtain CoP rate-of-change in seconds (dCoP).” Differentiating a position gives velocity – m/s and not s.

The reviewer is correct. “s” has been corrected to “m/s”.

13. line 219: “before and after at the” delete at

“at” was deleted.

14. Figure 3: x-axis labels are missing and x-axis should be given in seconds.

x-axis labels were added.

15. Line numbers are missing starting at page 14.

Line numbers have been added.

16. Page 14 bottom: weather should be whether

Word has been corrected.

17. Fig. 6: legend for different colors are missing.

Fig 6. Has been modified and legend was added.

18. Fig. 7 and 8: I suggest to rename “general muscle activity” by something like “EMG integral”, since the current label is very confusing.

As suggested we renamed “general muscle activity” to “EMG Integral”.

19. Page 17: the introduction of the discussion is much better in clarifying the hypotheses as compared to Abstract and Introduction. Maybe the latter could be reformulated along those lines.

We now introduce the hypotheses more clearly in the abstract and in the introduction and more along the line as in the introduction of the discussion (Line 39 – 34 and Line 161 – 166).

20. Page 22: “cuing or hether they” is this supposed to mean whether?

This word was supposed to mean whether. Word has been corrected.

21. Page 23: “asymptote, with a decrease of 15% in both the 1% and 4% force push condition and 20%” should be “1% and 7%”

There was a mistake with the percentages. This has been corrected. In order to make it easier understandable we changed the wording. 1% is now called small perturbation, 4% is called medium perturbation and 7% is called strong perturbation. Manuscript has been changed accordingly.

22. Page 31: “However, we did found an unexpected effect of cTBS stimulation in terms of improvements of the aforementioned adaptive process.” Should be “we did find”.

Wording has been corrected.

---

## [Decision Letter · Decision Letter 1]

26 Mar 2020

PONE-D-19-35562R1

Stabilization of body balance with light touch following a mechanical perturbation: Adaption of sway and disruption of right Posterior Parietal Cortex by cTBS

PLOS ONE

Dear Mr Kaulmann,

Thank you for submitting your manuscript to PLOS ONE. After careful consideration, we feel that it has merit but does not fully meet PLOS ONE’s publication criteria as it currently stands. Therefore, we invite you to submit a revised version of the manuscript that addresses the points raised during the review process.

Please pay particular attention to the comments of reviewer #1 regarding data analysis and interpretation. If the reviewer is still not satisfied with a revised version of the manuscript, the manuscript will be rejected.

We would appreciate receiving your revised manuscript by May 10 2020 11:59PM. To enhance the reproducibility of your results, we recommend that if applicable you deposit your laboratory protocols in protocols.io, where a protocol can be assigned its own identifier (DOI) such that it can be cited independently in the future. For instructions see: http://journals.plos.org/plosone/s/submission-guidelines#loc-laboratory-protocols

We look forward to receiving your revised manuscript.

Kind regards,

Andreas Kramer

Academic Editor

PLOS ONE

Reviewers' comments:

Reviewer's Responses to Questions

**Comments to the Author**

1. If the authors have adequately addressed your comments raised in a previous round of review and you feel that this manuscript is now acceptable for publication, you may indicate that here to bypass the “Comments to the Author” section, enter your conflict of interest statement in the “Confidential to Editor” section, and submit your "Accept" recommendation.

Reviewer #1: (No Response)

Reviewer #2: All comments have been addressed

2. Is the manuscript technically sound, and do the data support the conclusions?

Reviewer #1: Yes

Reviewer #2: Yes

3. Has the statistical analysis been performed appropriately and rigorously? 

Reviewer #1: No

Reviewer #2: Yes

4. Have the authors made all data underlying the findings in their manuscript fully available?

Reviewer #1: Yes

Reviewer #2: No

5. Is the manuscript presented in an intelligible fashion and written in standard English?

Reviewer #1: Yes

Reviewer #2: Yes

6. Review Comments to the Author

Reviewer #1: I thank the authors for taking into account my previous comments. I recognize that this is a complex study with a large amount of results to present, and it is not easy to do so. However, currently, it remains difficult for the reader to clearly understand the analyses performed and the results obtained. A particular attention must be given to the description of the models and of the description and display of the post hoc analyses.

-I was not aware of the k-mean procedure used by the authors. I find it very interesting. Are you confident that the k-means procedure can be reliable with only 11 subjects to cluster data in a relevant way? If you think so, maybe you could indicate why in the manuscript. Please detail this procedure in the paper: have you pooled all data together or have you repeated the procedure for each different variable? If it was the latter, were the 2 removed subjects clustered together for each variables or only a few? If the latter was the case, this may not be very convincing. To note: the shrinkage property of linear mixed models reduces the influence of outliers, so maybe you don’t really need to remove these subjects (but this is a personal observation).

-The statistical models are not sufficiently described by the authors. The authors said they have maximized the models but in the statistic section, it is written that the only random effect was “subject” (while one would expect random slopes and intercepts for each factor and interaction between factors). Similarly, in the result section you display interactions effects (e.g. Light Touch x Test), while these interaction effect are not described in the statistic section. Please indicate clearly the full model equations e.g. Dependant Variable ~ Factor1*Factor2*…+(Factor1*Factor2*…|Subject). Adding random effects for the interactions (as done in the model here) is important, otherwise the type I error rate may go through the roof (see Barr 2013). If you already did all that, please just aadd all the details in the adequate section.

-The post hoc analyses are not well described and results should be indicated on the corresponding figures to help the reader understand them. Given that you made observations per block, and that your initial models are also dependant of block levels, it seems fair to convey these post hoc analyses also per block level and not just pre vs post as you have done, otherwise there is no interest in using mixed models. But see my last comment before adressing this point (analysis should be done on the EMG normalized to baseline). As a side note, with the type of statistical analysis you have used, you can access model estimates with the summary(model) function. These estimates often make post hoc tests useless since they often answer the required questions, and more importantly for the present case, indicate from where the interaction stems. Displaying these estimates are in general much more informative than the p-values obtained with the function anova(model).

-I am not sure how you obtained your coefficients in table 4 given the models you have used. You have several fixed effects as factors and therefore you should have a coefficient per factor and per level (plus the interaction coefficients). You can display these coefficients with the function summary(model).

-I think it is a very good idea to normalize EMG values to baseline. Then, you have to redo the tests on the normalized data, which maybe have normal residuals without log transformation. In this case, post hoc analysis must focus on between condition difference at each block level (and please, indicate the results of these post hoc tests on the figures). If significant differences between conditions are observed in blocks performed before the cTBS treatment, you should definitely indicate to the reader in both the abstract and discussion that EMG data variability prevents any clear observation of the effect of cTBS on EMG. If this is the case, this could partly explain why the changes in EMG are not accompanied by behavioural changes during the balance task. If there is no difference in EMG pre cTBS, then you will have a much stronger support for your hypothesis.

Reviewer #2: All comments have been addressed appropriately.

The authors state that the data will be made available. I was not able to access at the time of the revision.

I only have a few minor corrections:

Line 49: “Furthermore, we saw gradual decrease of muscle activity,” upon what?

Line 269: some spaces are missing.

Line 562: reference style: [Park et al., 2004[40]]

7. PLOS authors have the option to publish the peer review history of their article (what does this mean?). If published, this will include your full peer review and any attached files.

Reviewer #1: No

Reviewer #2: Yes: Lorenz Assländer

---

## [Author Response · Author response to Decision Letter 1]

10 May 2020

Dear Editor,

we would like to thank the reviewers for their very valuable criticisms and comments regarding our study “Stabilization of body balance with light touch following a mechanical perturbation: Adaption of sway and disruption of right Posterior Parietal Cortex by cTBS”. We followed the reviewers’ advice and revised our manuscript thoroughly. Below we explain the changes made to the manuscript point-by-point.

Reviewer 1

1. I was not aware of the k-mean procedure used by the authors. I find it very interesting. Are you confident that the k-means procedure can be reliable with only 11 subjects to cluster data in a relevant way? If you think so, maybe you could indicate why in the manuscript. Please detail this procedure in the paper: have you pooled all data together or have you repeated the procedure for each different variable? If it was the latter, were the 2 removed subjects clustered together for each variables or only a few? If the latter was the case, this may not be very convincing. To note: the shrinkage property of linear mixed models reduces the influence of outliers, so maybe you don’t really need to remove these subjects (but this is a personal observation).

We decided for k-means clustering, because it is a simple but popular unsupervised machine learning algorithms, making sure data is clustered according to similarity once the number of clusters has been defined. Following the Reviewers suggestion, we provided more information regarding the procedure of cluster analysis. To answer the Reviewer question, data has been pooled together in order to analyses clusters (Line 267 – 274). 

2. The statistical models are not sufficiently described by the authors. The authors said they have maximized the models but in the statistic section, it is written that the only random effect was “subject” (while one would expect random slopes and intercepts for each factor and interaction between factors). Similarly, in the result section you display interactions effects (e.g. Light Touch x Test), while these interaction effect are not described in the statistic section. Please indicate clearly the full model equations e.g. Dependent Variable ~ Factor1*Factor2*…+(Factor1*Factor2*…|Subject). Adding random effects for the interactions (as done in the model here) is important, otherwise the type I error rate may go through the roof (see Barr 2013). If you already did all that, please just add all the details in the adequate section.

Full model equations have been added to the statistics section. However, it remains true that only subject is treated as a random factor. All other factors are treated as fixed effects. To our knowledge, this is a valid way of designing and building linear mixed model (Line 279 – 305). 

3. The post hoc analyses are not well described and results should be indicated on the corresponding figures to help the reader understand them. Given that you made observations per block, and that your initial models are also dependent of block levels, it seems fair to convey these post hoc analyses also per block level and not just pre vs post as you have done, otherwise there is no interest in using mixed models. But see my last comment before addressing this point (analysis should be done on the EMG normalized to baseline). As a side note, with the type of statistical analysis you have used, you can access model estimates with the summary(model) function. These estimates often make post hoc tests useless since they often answer the required questions, and more importantly for the present case, indicate from where the interaction stems. Displaying these estimates are in general much more informative than the p-values obtained with the function anova (model).

There seems to be a small confusion regarding our models. Our initial model is based on pre vs. post and not block. That is also the reason we decided to add a post hoc test, in order to clarify the effects especially of the stimulation protocol. We tried to describe the post hoc analyses in a more concise and clear matter, in order to make this connection clearer. Only in our second model did we analyze the progression over time (block). However, also for this model did we add a post hoc test, in order to investigate the block differences, especially in the first 4 blocks, as the Reviewer suggested. We did not report model estimates at first, because in our opinion it would flood the manuscript with too many tables. However, due to the Reviewer’s comments we decided to add the model estimates to the appendix, in order to make them available for those interested (Line 300 – 304). 

4. I am not sure how you obtained your coefficients in table 4 given the models you have used. You have several fixed effects as factors and therefore you should have a coefficient per factor and per level (plus the interaction coefficients). You can display these coefficients with the function summary(model).

Thanks to the Reviewers comment we realized that this coefficient table was a left over from a previous analysis that we excluded in the final version of manuscript. This table should have been removed as well. We removed the table form the manuscript, since we did not discuss nor reference it anymore and it became obsolete.

5. I think it is a very good idea to normalize EMG values to baseline. Then, you have to redo the tests on the normalized data, which maybe have normal residuals without log transformation. In this case, post hoc analysis must focus on between condition difference at each block level (and please, indicate the results of these post hoc tests on the figures). If significant differences between conditions are observed in blocks performed before the cTBS treatment, you should definitely indicate to the reader in both the abstract and discussion that EMG data variability prevents any clear observation of the effect of cTBS on EMG. If this is the case, this could partly explain why the changes in EMG are not accompanied by behavioral changes during the balance task. If there is no difference in EMG pre cTBS, then you will have a much stronger support for your hypothesis.

We analyzed the data with the normalized EMG values as suggested by the Reviewer. Effects of the stimulation protocol for the Gastrocnemius disappeared but became stronger and clearer for the Tibialis Anterior. We changed the tables and figures accordingly. As the reviewer suggested we also looked at the block differences, specifically in the time before stimulation. We added the post hoc test to the statistics and result section. Even though we find effects of block and touch, no differences in regard to stimulation session are present. 

Reviewer 1

1. The authors state that the data will be made available. I was not able to access at the time of the revision.

Data will be made available once the Manuscript has been published, as it is stated in the data availability policy. 

2. Line 49: “Furthermore, we saw gradual decrease of muscle activity,” upon what?

Sentence has been improved with additional information (Line 50).

3. Line 269: some spaces are missing.

Spaces have been added

4. Line 562: reference style: [Park et al., 2004[40]]

Reference style has been corrected.

---

## [Decision Letter · Decision Letter 2]

18 May 2020

Stabilization of body balance with light touch following a mechanical perturbation: Adaption of sway and disruption of right Posterior Parietal Cortex by cTBS

PONE-D-19-35562R2

Dear Dr. Kaulmann,

We are pleased to inform you that your manuscript has been judged scientifically suitable for publication and will be formally accepted for publication once it complies with all outstanding technical requirements.

With kind regards,

Andreas Kramer

Academic Editor

PLOS ONE

Additional Editor Comments (optional):

Reviewers' comments:

Reviewer's Responses to Questions

**Comments to the Author**

1. If the authors have adequately addressed your comments raised in a previous round of review and you feel that this manuscript is now acceptable for publication, you may indicate that here to bypass the “Comments to the Author” section, enter your conflict of interest statement in the “Confidential to Editor” section, and submit your "Accept" recommendation.

Reviewer #1: All comments have been addressed

Reviewer #2: All comments have been addressed

2. Is the manuscript technically sound, and do the data support the conclusions?

Reviewer #1: Yes

Reviewer #2: Yes

3. Has the statistical analysis been performed appropriately and rigorously? 

Reviewer #1: Yes

Reviewer #2: Yes

4. Have the authors made all data underlying the findings in their manuscript fully available?

Reviewer #1: Yes

Reviewer #2: Yes

5. Is the manuscript presented in an intelligible fashion and written in standard English?

Reviewer #1: Yes

Reviewer #2: Yes

6. Review Comments to the Author

Reviewer #1: I thank the authors for answering all my comments, and for the added material in the methods and supplementary material section.

Reviewer #2: (No Response)

7. PLOS authors have the option to publish the peer review history of their article (what does this mean?). If published, this will include your full peer review and any attached files.

Reviewer #1: No

Reviewer #2: Yes: Lorenz Assländer

---

## [Editor Report · Acceptance letter]

17 Jun 2020

PONE-D-19-35562R2 

Stabilization of body balance with light touch following a mechanical perturbation: Adaption of sway and disruption of right Posterior Parietal Cortex by cTBS 

Dear Dr. Kaulmann:

I'm pleased to inform you that your manuscript has been deemed suitable for publication in PLOS ONE. Congratulations! Your manuscript is now with our production department. 

Kind regards, 

on behalf of

Dr. Andreas Kramer 

Academic Editor

PLOS ONE